# Automated Pavement Construction Inspection Using Uncrewed Aerial Systems (UAS)—Hot Mixed Asphalt (HMA) Temperature Segregation

**Reihaneh Samsami** [1,*]**, Amlan Mukherjee** [2] **and Colin N. Brooks** [3]

1   Department of Civil and Environmental Engineering, University of New Haven, West Haven, CT 06516, USA
2   Department of Civil, Environmental, and Geospatial Engineering, Michigan Technological University, Houghton, MI 49931, USA; amlan@mtu.edu
3   Michigan Tech Research Institute (MTRI), Michigan Technological University, Houghton, MI 49931, USA; cnbrooks@mtu.edu
*   Correspondence: rsamsami@newhaven.edu; Tel.: +1-9063705312

**Abstract:** Temperature segregation in Hot Mixed Asphalt (HMA) pavement construction leads to performance problems, such as reduced fatigue life. During construction, Quality Assurance (QA) inspection procedures are required to evaluate the pavement condition and detect the segregated areas. In traditional HMA highway construction inspection processes, temperature differences are investigated manually, by sampling the HMA behind the paver. In these processes, inspectors are required to work adjacent to traffic and alongside moving or backing equipment. These processes do not provide a complete temperature profile of the mat, endanger the inspectors' safety, and require on-site experienced inspectors. An Uncrewed Aerial System (UAS) enables HMA pavement construction inspection to be conducted within a remote, non-destructive, safe, and efficient framework. The objective of this research is to design an automated UAS imaging workflow for HMA pavement construction inspection, mainly locating temperature segregation. The primary contribution of this paper is to provide Departments of Transportations (DOTs) and contractors with workflows for creating enhanced remote inspection procedures and detailed thermal profiles of the placed HMA mat. The application of the proposed workflow is illustrated using an HMA construction project in Michigan.

**Keywords:** automated inspection; Uncrewed Aerial System (UAS); highway construction inspection; Hot Mixed Asphalt (HMA) pavement; Quality Assurance (QA); thermal segregation; thermal profiling





## 1. Introduction

Segregation, defined as "the non-uniform distribution of coarse and fine aggregate components within the asphalt mixture", is a common problem in Hot Mixed Asphalt (HMA) pavement construction [1]. Several performance problems such as reduced stiffness, reduced fatigue life, reduced tensile strength, and increased permeability are caused by segregation [2,3]. There are two types of segregation, namely, "gradation segregation" and "temperature segregation". As defined by the American Association of State Highway and Transportation Officials (AASHTO), gradation segregation is caused by "concentration of coarse material in some areas of the paved mat, while other areas contain a concentration of finer material" [4,5]. Temperature gradation is defined as "non-uniform temperature distribution across the mat of uncompacted asphalt mixtures during paving operation" [6]. While the cause of and cures for gradation segregation are widely studied in the literature, there is limited investigation on temperature segregation detection and the development of automated workflows for investigating it. Consequently, this paper focuses on detecting temperature segregation in HMA construction, using new tools and technologies.

In the current state of inspection practices, inspectors refer to Quality Assurance (QA) instructions and HMA production manuals to measure the temperature differentials in the freshly placed pavement. For instance, Michigan Department of Transportation (MDOT) recommends referring to MTM 324-07 test instructions for "Sampling HMA Paving Mixtures Behind the Paver" to manually collect thermal data, immediately behind the paver and before the initial compaction [7]. This requires inspectors to work adjacent to traffic and alongside moving or backing equipment. These processes endanger the inspectors' safety and require on-site experienced inspectors. In addition, the manual inspection data acquisition and data extraction from project documents is labor intensive, requiring extensive time and calculations [8]. Therefore, there is an opportunity for safer, accurate, and more efficient thermal data collection practices for HMA construction inspection.

Thermal imaging is a method for identifying temperature differentials and locating "temperature segregation" in asphalt. Pioneered by Read [9] at the Washington State Department of Transportation (WSDOT), several research projects have been conducted on determining temperature differences in HMA and relating them to segregation [1,2,9]. Thermal images can be taken using different tools, such as paver-mounted thermal sensors or an Uncrewed Aerial System (UAS) equipped with thermal sensors. While this research focuses on using a UAS as a data collection tool, the proposed methodology can be applied to any other tool.

An Uncrewed Aerial System (UAS) is one of the tools that has the potential to support automation of the HMA inspection process through the construction phase. It is a system consisting of an Uncrewed Aerial Vehicle (UAV), ground control station, and multiple sensors. By mounting a thermal camera on the UAS and flying it over the HMA construction operation, a thermal profile of the HMA mat being placed can be developed.

The objective of this research is to propose an automated inspection workflow for highway construction projects. The primary contribution of this paper is to provide transportation agencies and contractors with workflows for creating enhanced remote inspection procedures and detailed thermal profiles of the placed HMA mat.

This paper is structured in six sections. The Introduction section is followed by a Literature Review section, where recent research in HMA thermal segregation is studied. This section also reviews the current state of practice for HMA manual inspection in Michigan, U.S. Section Three elaborates the research methodology in the following three steps: (a) data collection, (b) data processing, and (c) data analysis and result. The proposed methodology is illustrated in Section Four, using an example HMA project by MDOT. The developed workflow is later discussed in Section Five, followed by a conclusion in the last section. The limitations of the workflow, as well as the direction for future research, are also explained in that section.

## 2. Literature Review

IR cameras have been identified as a reliable method for segregation detection in an HMA mat [3]. Brock and Jakob [10] are credited with initial use of IR camera images to create temperature differential records of HMA paving operations. There are several research projects, mostly funded by DOTs, exploring temperature differential ranges and relating them to asphalt segregation. For instance, Stroup-Gardiner and Brown [3] reported temperature differentials of more than 19 °F (10 °C) as potentially segregated areas and temperature differentials of more than 36 °F (20 °C) as highly segregated areas. As a result, the Arkansas DOT is limiting the "behind the paver" thermal differences to less than 19 °F (10 °C) to avoid asphalt segregation [11].

Several research studies investigated the compaction characteristics of HMA under different temperatures [12–14]. They concluded that compaction must be completed before the inner temperature of the HMA mat decreases to cessation temperature (below 185 °F (85 °C) or 176 °F (80 °C)). For temperatures below this limit, HMA becomes relativity rigid, and compaction results in an increase in air voids.

In addition, other studies [13,14] have focused on the ideal "temperature window" for HMA compaction. According to their study, the temperature window for HMA compaction is 176–293 °F (80–145 °C). For instance, Hainin et al. [13] worked on density profiling of HMA for various roller types and lift thicknesses. According to their investigation, extra rolling may result in decompaction for mix temperatures below 80 °C. Starting compaction in the temperature range of 284–302 °F (140–150 °C) results in a pavement with high resistance against cracking. Compacting outside the temperature window also resulted in 30% reduced cracking toughness despite compacting to the desired density.

Kim et al. [15] also studied the effect of temperature segregation on achieving desired mechanical properties and density of HMA. As stated by those authors, air voids in the compacted layer increase, and the density decreased when the laydown temperature loss was greater than 50 °F (10 °C). This happens when there is a considerable delay in starting the compaction following the HMA laydown.

Consistent paving and proper rolling practices are also another field of study, as discussed in Arbeider et al. [16]. According to their study, to reach consistency in paving, the roller speed should be equal to the paver speed to eliminate any lags between HMA laydown and compaction. These lags are the main reason for HMA mat inconsistencies.

Despite the broad existing literature on collecting thermal data using traditional and infrared (IR) handheld guns/cameras, there is limited research on using UAS-mounted thermal cameras for thermal profiling purposes.

While several recent studies [17–28] have investigated the application of a UAS on pavement distress detection, there is limited research on utilizing a UAS for HMA thermal segregation inspection. For instance, Du et al. [23] reviewed the application of digital image processing tools on pavement distress detection, including thermal segregation. In their study, they have reviewed several data acquisition methods such as (a) cameras, (b) thermal imaging technologies, (c) laser, and (d) ground generation radar. Those authors have also made a comparison between different imaging tools and reviewed their advantages and limitations. Some of the benefits of using thermal imaging technologies are stated as convenient portability, contactless testing, absence of harmful radiation, and the ability to capture real-time images.

In addition, there is no automated workflow proposed for HMA construction inspection, using thermal images. To address these two gaps, the applications and limitations of using a UAS in an automated construction inspection workflow must be studied. In addition, the potential beneficiaries of this method, including inspectors, DOTs, agencies, and contractors should be identified. Therefore, this paper reviews the application of a UAS for automated HMA pavement construction inspection, by illustrating a case study of the I-69 MDOT HMA construction project. In collaboration with Michigan Tech Research Institute (MTRI), the authors have created a thermal visualization of an ongoing HMA pavement project in Lansing, Michigan. They have also introduced thermal segregation metrics and estimated them for the I-69 project.

*Current State of Practice—MDOT*

The HMA production manual (2020) provides the standard operational guidelines for achieving a quality product [7]. Its target audience is engineering, technical, and administrative staff. The four sections of this manual are: (1) Procedures Manual for Processing HMA Mix Design, (2) Certification Procedures of HMA Plants, (3) HMA Quality Control/Quality Assurance Procedures for Field Testing, and (4) HMA Laboratory and Technician Qualification Program. The third section provides guidance on in-field QA testing of the HMA. For the purpose of this study, the HMA temperature-related QA subsection is reviewed.

MTM 324 "Sampling Behind the Paver" and MTM 313 "Sampling HMA Loose Mix from Mini-stockpile" are the two Michigan Test Methods (MTM) accepted by MDOT for the purpose of temperature testing. In MTM 324, samples are taken immediately behind the paver and before the initial roller compaction [7]. Figure 1 illustrates the pattern the

samples need to be taken. As shown by black circles in this Figure, each composite sample consists of a minimum of three samples taken within 10 ft of a longitudinal distance and across the width of the pavement.

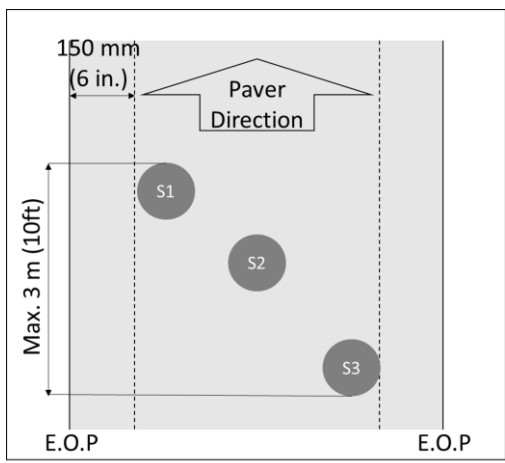

**Figure 1.** Sampling pattern behind the paver [7].

In Figure 2, the manual "Sampling HMA Paving Mixtures Behind the Paver" is illustrated. As shown, the inspector uses sampling plates and wires, a non-absorbent container, an MDOT approved shovel, and an MDOT-approved splitter to conduct the test (Figure 2A). A minimum of three samples are taken immediately behind the paver, at random locations. For this purpose, plates with wire leads are placed ahead of the paver, and the wire lead is extended beyond the edge of pavement. The wired leads are used to locate the wire leads (Figure 2B) and then dig the pavement above the plate, using shovel. This material is placed in the container, and the voids are filled/levelled by the contractor afterwards (Figure 2C).

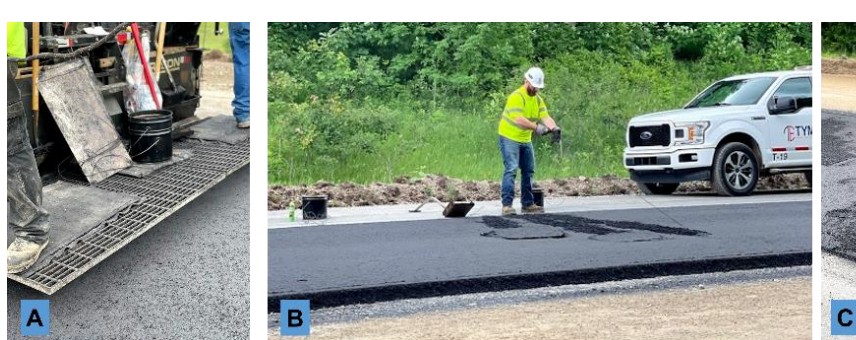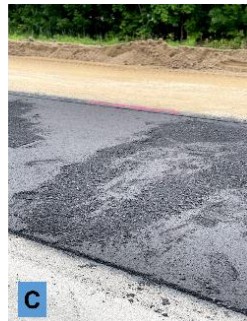

**Figure 2.** MTM 324 inspection for I-69 project (**A**: Sampling plates and wires, **B**: Taking samples, **C**: Filling the voids).

There are several inspection criteria for inspecting the HMA temperature at different stages of an HMA pavement process. These criteria are broken down by MDOT into (a) General Inspection, (b) HMA Preparation, (c) Pavement Operation, and (d) Compaction, Coring, etc. Based on these criteria, the temperature-related inspection items are:

1.  Wedging must be cooled to support construction equipment without causing visible distortion of the mat before placing subsequent wedging, base, leveling, or top course mixtures.
2.  If delays slow paving operations and the mat temperature immediately behind the screed falls below 200 °F (93 °C), then paving must be stopped, and a transverse construction joint must be placed.
3.  If the temperature of the mat falls below 190 °F (88 °C) before initial breakdown rolling, the mat must be removed.

4.   If the temperature of the mat falls below 170 °F (77 °C), a bond coat must be applied to the vertical edge of the mat, before placing the adjacent mat.

5.   When constructing lanes with at least two pavers in echelon, the depth of loose HMA from each paver at the longitudinal joints must be matched.

Building on these MDOT HMA inspection items and on the existing literature, a set of thermal segregation parameters is introduced for the HMA placement and compaction operations in Table 1. These parameters will be estimated using the proposed methodology, to evaluate the segregated areas.

**Table 1.** HMA quality inspection thermal segregation parameters.

| No | Parameter | Criterion | Note |
|---|---|---|---|
| 1 | $R_c$ | Temperature Range | Is laying temperature within the temperature window of 185 °F–293 °F (85–145 °C) considered? |
| 2 | $R_{ps}$ | Temperature Differentials 1 | Are temperature differentials larger than 19 °F (10 °C) causing potentially segregated spots? |
| 3 | $R_{hs}$ | Temperature Differentials 2 | Are temperature differentials larger than 36 °F (20 °C) causing highly segregated spots? |

It must be noted that each DOT has its own HMA QA procedures, and the parameters in Table 1 must be modified for application by other DOTs. For example, the lower limit for temperature range is 165 °F (74 °C) for KSDOT [29]. Another example is Caltrans, where several criteria are offered as minimum compaction temperatures as 250 °F (121 °C) for first coverage of breakdown compaction, 190 °F (88 °C) for breakdown and intermediate compaction, and 150 °F (66 °C) for finish compaction for HMA with unmodified binder. For HMA with modified binder, the minimum temperatures are 10 °F (5 °C) lower than these temperatures [30].

### 3. Methodology

In this section, an automated inspection methodology is proposed and explained in three subsections of (a) data collection, (b) data processing, and (c) data analysis and result. Table 2 summarizes the proposed workflow for HMA construction inspection. It illustrates how different data is collected from HMA pavement activity, and what information products are created in each phase of the project. It also identifies potential stakeholders interested in the metrics created at the end of the data analysis process. Using these metrics, the identified stakeholders will be able to accept the project or compare as-planned quantities with as-built data for progress monitoring purposes.

**Table 2.** Proposed automated inspection workflow.

| Construction Site Activity | Data Collected | Information Products | Project Phase Impacted | Interested Stakeholder | Metric or Process | Decision Informed |
|---|---|---|---|---|---|---|
| **HMA Pavement** | Thermal Data | Cool Spots (Geo-TIFF & JPG) | Construction | Inspector Contractor DOT | Quality Control Thermal Segregation | Project Acceptance |
| | Visual Data | As-Built BIM (Civil 3D & Microstation) | Performance & Monitoring | DOT | Integration with Asset Management Visualization | As-Planned vs As-Built Comparison |

#### 3.1. Data Collection

Figure 3 captures the thermal data collection by MTRI at I-69 project. Before flying, the process starts with placing ground control points (GCPs) (Figure 3A) to provide high-accuracy georeferencing information for the collected imagery. The GCPs used here were

Aeropoint units from Propeller, which provided approximately 3 cm positional accuracy for the center of each target using built-in GPS and post-processing capabilities. This positional information improves the output quality of registered and merged images.

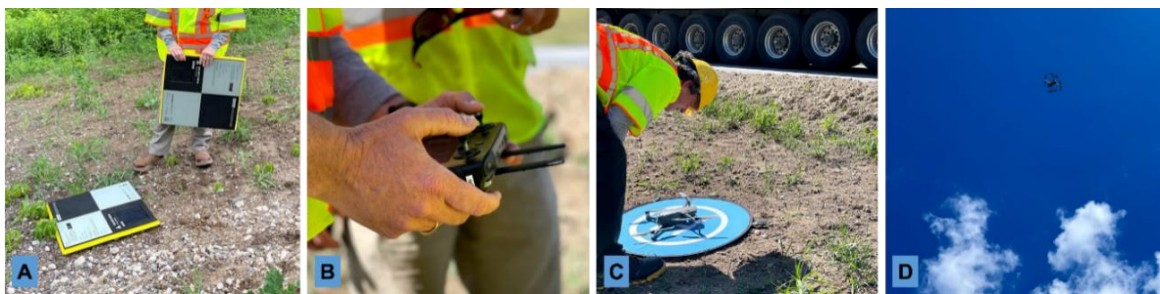

**Figure 3.** UAS thermal imaging set up on I-69 (**A**: Placing GCPs, **B**,**C**: Pilot flying the UAS, **D**: UAS flying at about 24m (80 ft)).

For the imaging part of the UAS framework, missions were pre-planned for the areas of interest using the DJI Pilot application, which is available on the DJI Smart Controller. The Pilot app includes the ability to set missions that are focused on obtaining a dense network of overlapping thermal images, with 80% front and side overlap, in case a thermal orthophoto is needed. Heights can vary, but typically, 24 m (80 feet) was useful for thermal missions. Optical (natural color) and thermal images were taken at the maximum rate of two frames per second. The boundaries of the mission area were drawn on the interactive Pilot app screen, with the thermal mission option, normal flying speeds around 4.2 kph (2.6 mph), and a 24 m flying height. Caution was taken to make sure that the UAS would not be flying over any moving traffic. The pilot in command (PIC) was ready to take control from the automated mission at all times.

Weather conditions were checked ahead of each flight starting 48 h ahead of time, using the U.S. National Weather Service and smart phone apps such as Windy and UAV Forecast, and then on the day of the data collection, including up to right before takeoff. Generally, the PIC would plan for collecting in steady wind speeds of no more than 24 kph (15 mph) with wind gusts below 40 kph (25 mph).

The M2EA UAS has the Multiple Display Mode technology to display both infrared and visible light images while collecting data. Figure 4 is an example of the data collected in Multiple Display Mode (infrared and visible) and then stored as a side-by-side comparison image when using this UAS. Thermal images recorded by the M2EA are 640 × 512 pixels and can be converted to per-pixel temperature values as this is a radiometric thermal camera. Visible light images are collected with a quad-Bayer filter camera that interpolates 12 mp images into 48 mp outputs. The already-paved shoulder and mainline (shown in yellow) is compared with the existing pavement (shown in orange/red) in the captured thermal and visible light images.

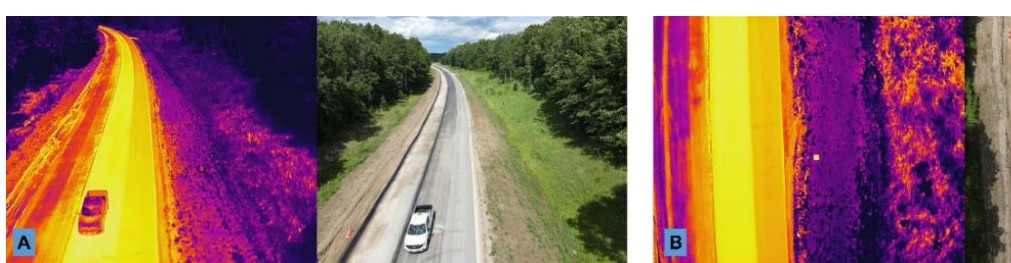

**Figure 4.** Thermal profile of already-paved shoulder/mainline vs. existing pavement (**A**: Side View Images, **B**: Top-Down Images).

### 3.2. Data Processing

Image processing is used as a tool to automatically identify the thermal differences in the HMA mat. Thermal images in the RJPG format are the output of the data collection procedure. Each image is a collection of pixels with three red, green, and blue intensities (IR, IG, and IB) that vary between 0 and 255. The first step is to process these images in the DJI Thermal Analysis Tool 2 application and create grayscale images. In a grayscale image, IR = IG = IB = I. Figure 5 is an example of this grayscale conversion for image DJI_0040.

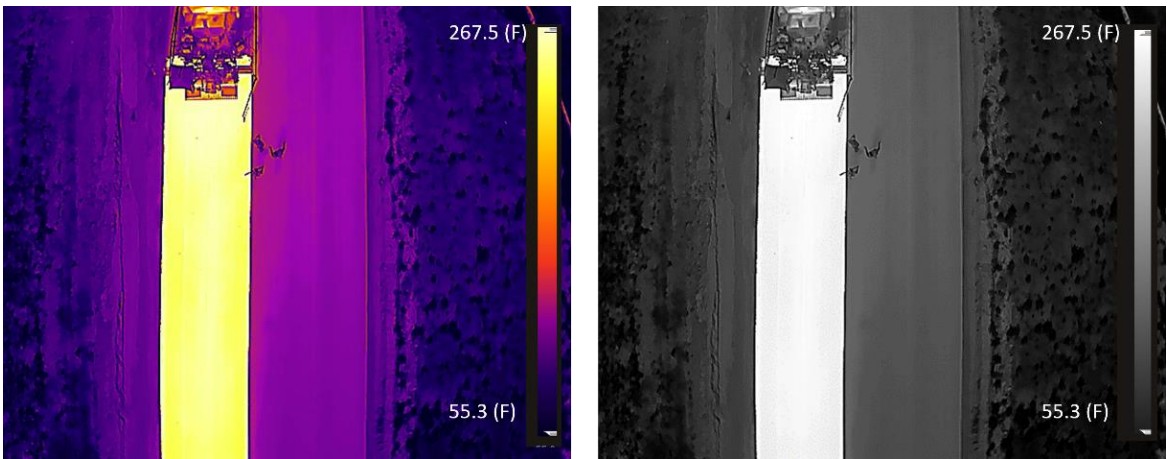

**Figure 5.** Grayscale image from regular RJPG.

The second step is to identify the thermal ranges on each image and associated color intensities. This requires using the DJI Thermal Analysis Tool 2 application to find the maximum and minimum temperatures on each grayscale image. Figure 6 captures this comparison between the two DJI_0049 and DJI_0040 images. Temperature ranges are (64.5 °F, 263.6 °F) and (55.3 °F, 267.5 °F) for these two images, respectively. As illustrated, the coldest spot on each image is associated with I = 0 (black), and the warmest spot with I = 255 (white). Different temperature ranges on each image make it difficult to compare images with each other.

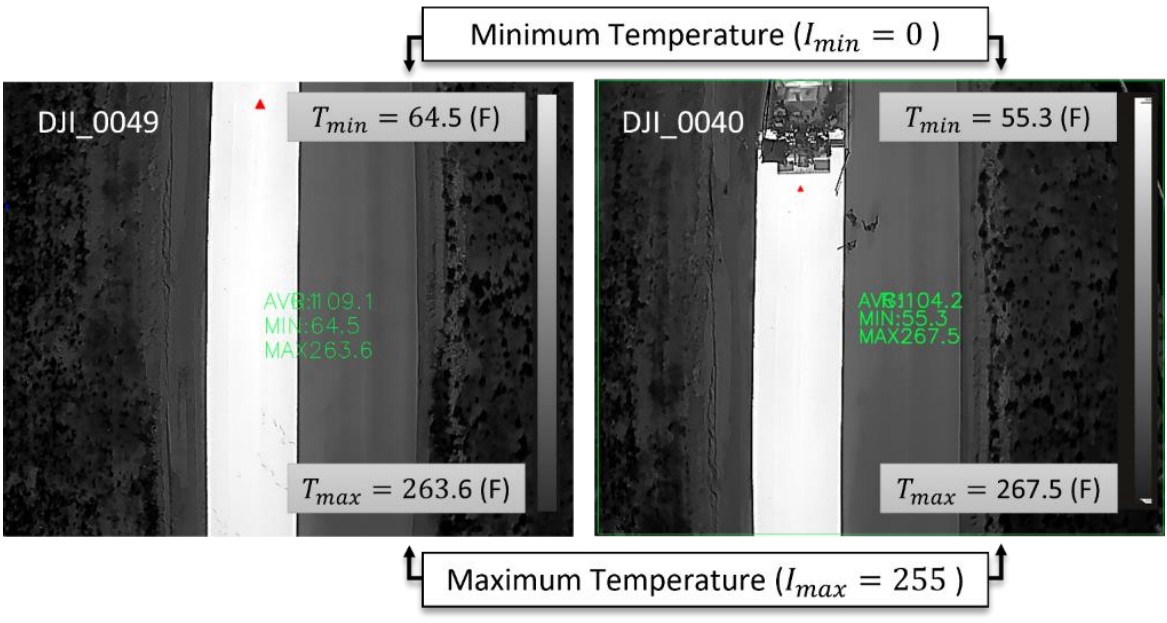

**Figure 6.** Comparison between temperature ranges (Red triangle locates the maximum temperature on each pavement image).

Therefore, the third step is to define a universal range and correct intensities considering that range (Equation (1)). All images are studied to find the minimum ($T_{min}$) and maximum temperatures ($T_{max}$). The highest maximum temperature among all images ($T'_{max}$ = Max ($T_{max}$)) is defined as the highest limit of the universal range. The lowest minimum temperature among all images ($T'_{min}$ = Min ($T_{min}$)) is defined as lowest limit of the universal range. The universal range is defined as (0 °F, 355 °F) for this set of collected data. It can be different for other data sets, based on the $T'_{max}$ and $T'_{min}$. Figure 7 illustrates the intensity–temperature graphs for DJI_0040 (as an example) and the newly defined universal range.

$$T = \frac{I'}{m\prime} + T'_{min} \quad Where \; m\prime = \frac{255 - 0}{355 - 0} \tag{1}$$

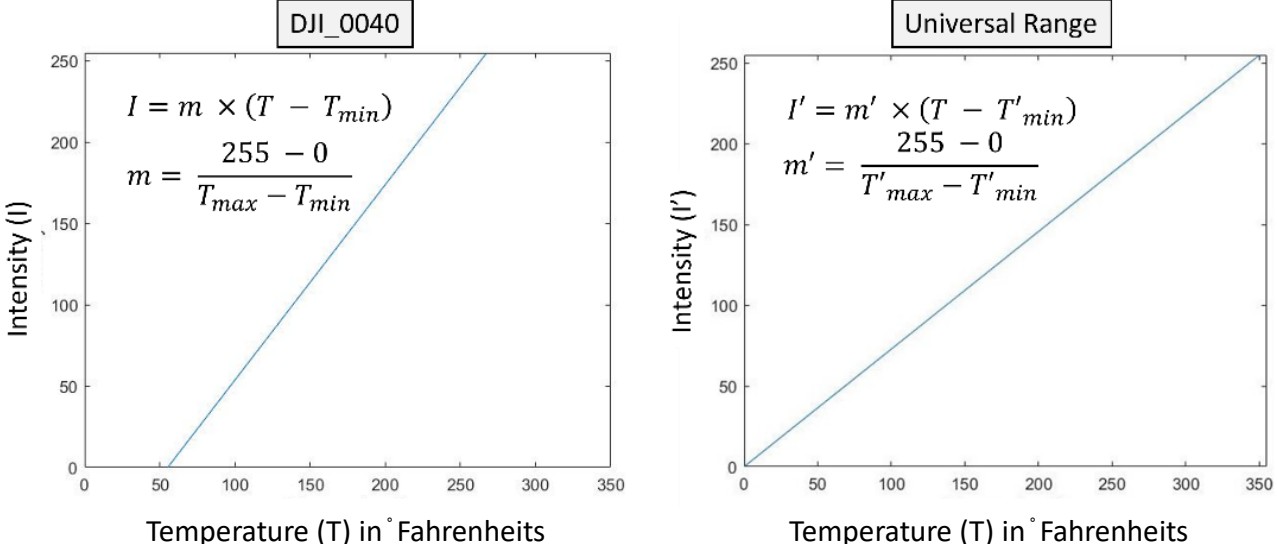

**Figure 7.** Intensity–temperature relationship for DJI_0040 and universal range.

### 3.3. Data Analysis and Results

After processing the images and converting them using the universal scale, they can be studied closely to estimate the thermal parameters and compare them with the manual inspection results. For this purpose, Equations below are used to estimate the three parameters in Table 1.

Equation (2) is used to estimate the ratio of cool areas. To recall, (185 °F, 293 °F) was defined as the temperature window for HMA pavement. Cool areas are defined as any HMA mat areas with temperatures below 185 °F. Depending on the project specifications, this lower limit can be manipulated to accurately estimate the cool areas in an HMA construction operation.

Equations (3) and (4) are used to estimate the potentially and highly segregated areas, respectively. A temperature difference of 19 °F is the maximum tolerable temperature difference on HMA mat before segregation happens. Areas with thermal differences between (19 °F, 36 °F) are identified as potentially segregated areas. Any mat area having thermal difference larger than 36 °F is identified as a highly segregated area. These numbers can also be customized based on the project specifications.

$$R_c = \frac{Number \; of \; Pixels \; with \; Temperatures \; under \; 185 \; °F}{Total \; Number \; of \; HMA \; Mat \; Pixels} \tag{2}$$

$$R_{ps} = \frac{Number \; of \; Pixels \; with \; Temperatures \; Differentials \; within \; (19 \; °F \; 36 \; °F)}{Total \; Number \; of \; HMA \; Mat \; Pixels} \tag{3}$$

$$R_{hs} = \frac{Number\ of\ Pixels\ with\ Temperatures\ Differentials\ greater\ than\ 36\ °F}{Total\ Number\ of\ HMA\ Mat\ Pixels} \qquad (4)$$

## 4. Application: I-69 MDOT Project

To illustrate the proposed workflow, Michigan DOT's reconstruction project of I-69, located in South Lansing, Ainger Road, was studied. Table 3 provides the information for this project for the I-69 MDOT project case study area. More information about this project has been posted at [31,32]. The project has focused on extending the life of the existing infrastructure.

**Table 3.** Information on MDOT I-69 rebuilding project area used for case study.

| Project Delivery Method | Design—Build |
|---|---|
| Owner | MDOT |
| Contractor | Michigan Paving |
| Inspector | HNTB |
| Contract Type | Alternate Bid |
| Mix Design | 3E30 Mix (Lansing Plant—Mainlines) |
| Project Duration | September 2020 to November 2023 |
| Length of Rebuilding Area | 40 km (25 miles) |

### 4.1. Data Collection

The authors visited the site on two data collection trips in July and October 2021. Table 4 summarizes the number/type of data collected in RJPG (Thermal data embedded in the JPG) and regular JPG formats.

**Table 4.** I-69 data collection information.

| Date | Flight | Data Collected | | Note |
|---|---|---|---|---|
| | | **Thermal (RJPG)** | **Visual (JPG)** | |
| 21 June 2021 | 100MEDIA | 10 | 10 | Test Flight. |
| | 101MEDIA | 333 | 333 | - |
| | 102MEDIA | 334 | 334 | - |
| | 103MEDIA | 99 | 99 | - |
| 10 June 21 | 104MEDIA | 133 | 133 | Paving had stopped. |
| | 105MEDIA | 333 | 333 | Paving had stopped. |
| | 106MEDIA | 333 | 333 | - |
| | 107MEDIA | 246 | 246 | - |

During site visits, the authors had an opportunity to become familiarized with the ongoing paving operation and shadow the project inspectors to understand the current state of practice. Figure 8 illustrates the HMA paving operation, where the HMA is transferred from truck to paver, using a Material Transfer Vehicle (MTV).

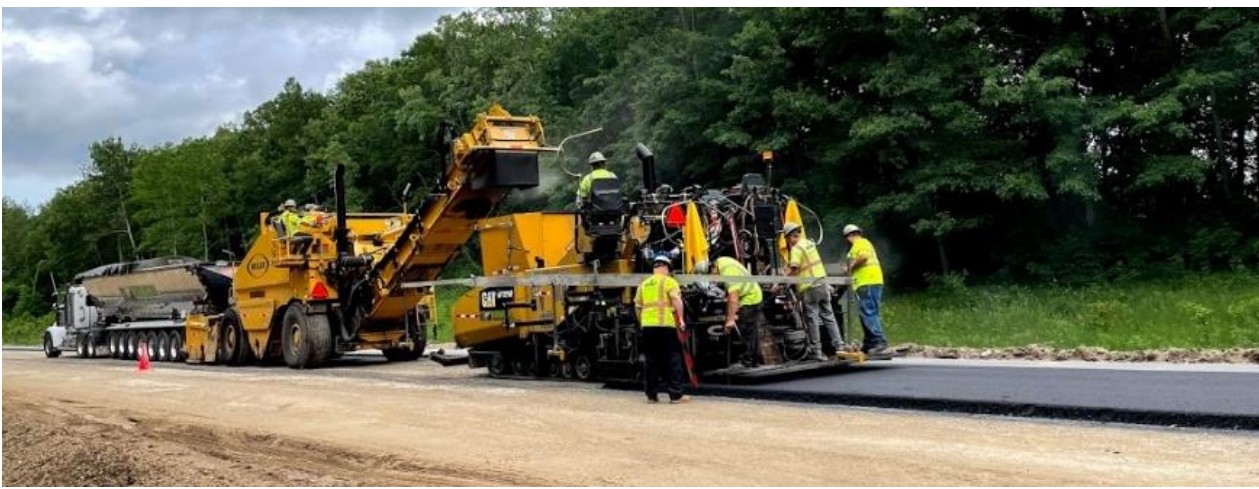

**Figure 8.** I-69 HMA pavement construction operation.

Figure 9 is an example of the dual images collected from the HMA in the hopper (Figure 9A) and HMA behind the paver (Figure 9B). Figure 10 is also an example of the dual image collected from the freshly placed HMA at the very beginning of the paved mat (Figure 10A) and along the roller tracks (Figure 10B). In these images, cold areas are shown in blue/purple hues, while warmer areas are in orange/yellow hues. For example, the HMA in the hopper has a high temperature and is shown in yellow. The existing pavement has a lower temperature and is shown in purple.

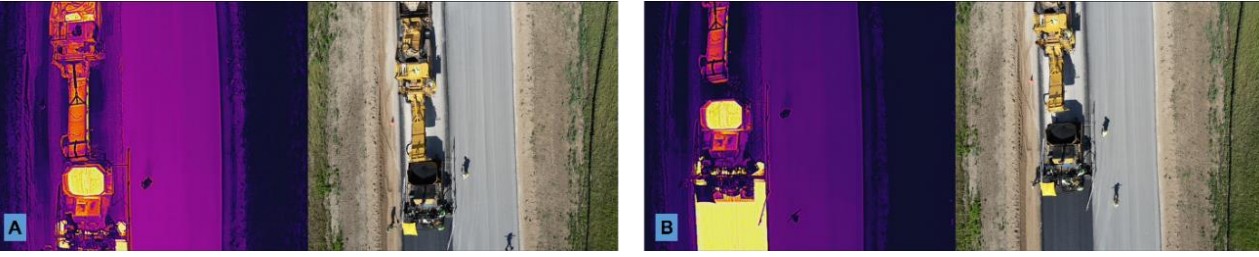

**Figure 9.** Thermal profile of HMA in the hopper (**A**) and behind the paver (**B**).

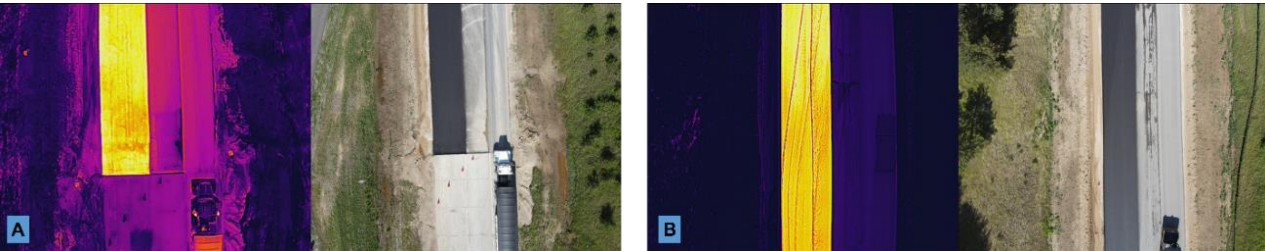

**Figure 10.** Differential cooling on the freshly placed HMA at the very beginning of the paving (**A**) and along the roller tracks (**B**).

*4.2. Data Processing*

Eight images are selected from the data pool (containing 2000 images) to investigate the proposed methodology. These images captured the newly placed HMA mat immediately behind the paver, at different locations. They are processed through the steps discussed in the Methodology section (Figure 11). Figure 12 presents these selected images, after they are scaled to the universal scale and contoured. To understand the contours better, consider the color bar in Figure 13. This color bar maps colors to temperature values in

the range (0 °F, 350 °F). The low temperatures in this range are shown in a blue color, while the moderate temperatures are shown in green-yellow, and the high temperatures are presented by red. As illustrated in Figure 12, the mat temperature is generally in the range of (200 °F, 300 °F), while the daylighting and the unpaved shoulder are in the range of (20 °F, 100 °F). There were some points on the collected images with extremely low or high temperatures. These points were resulted from cooling/heating system components of the equipment in the images. There are no such extreme points in any of the selected images in Figure 12.

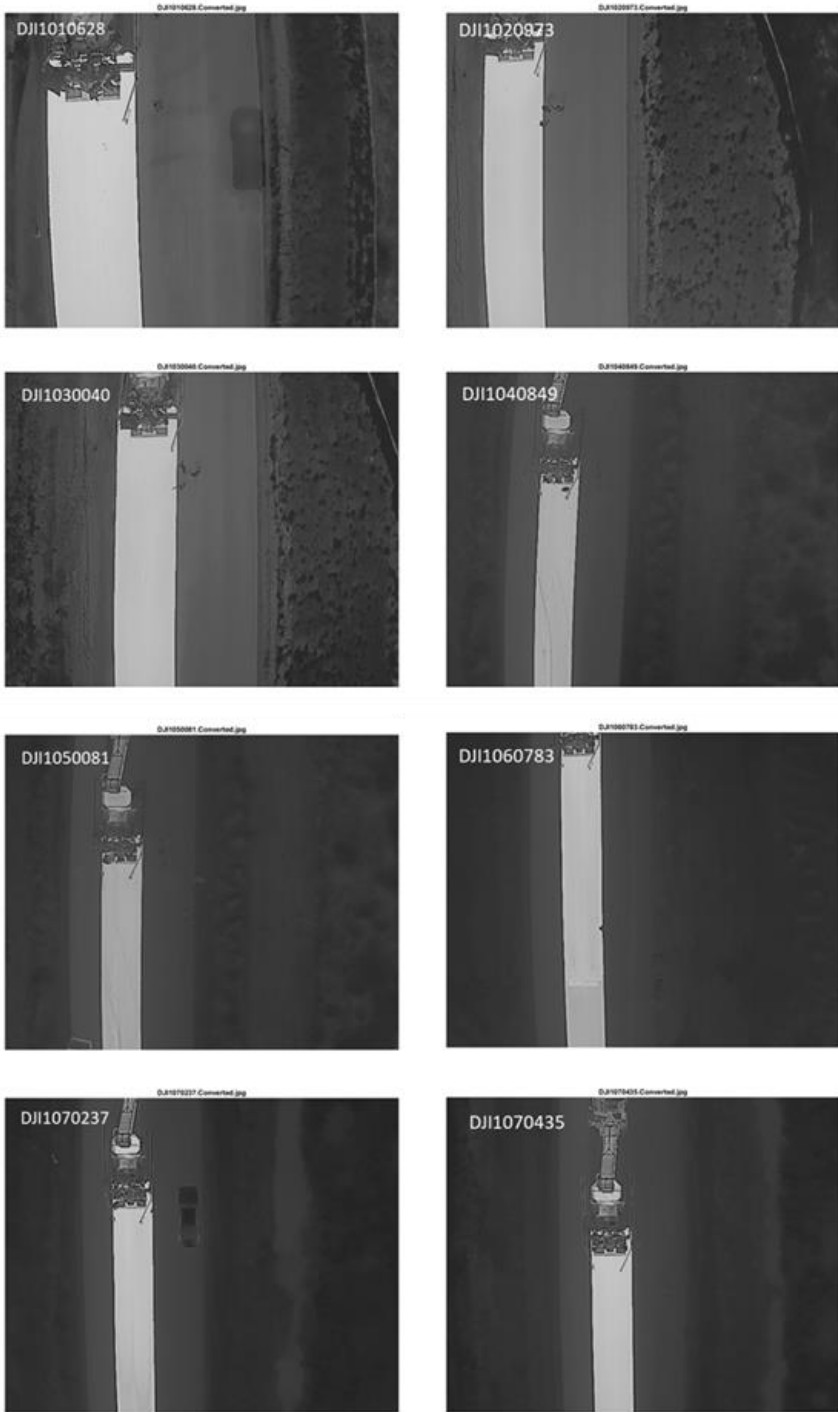

**Figure 11.** Selected images after scale conversion.

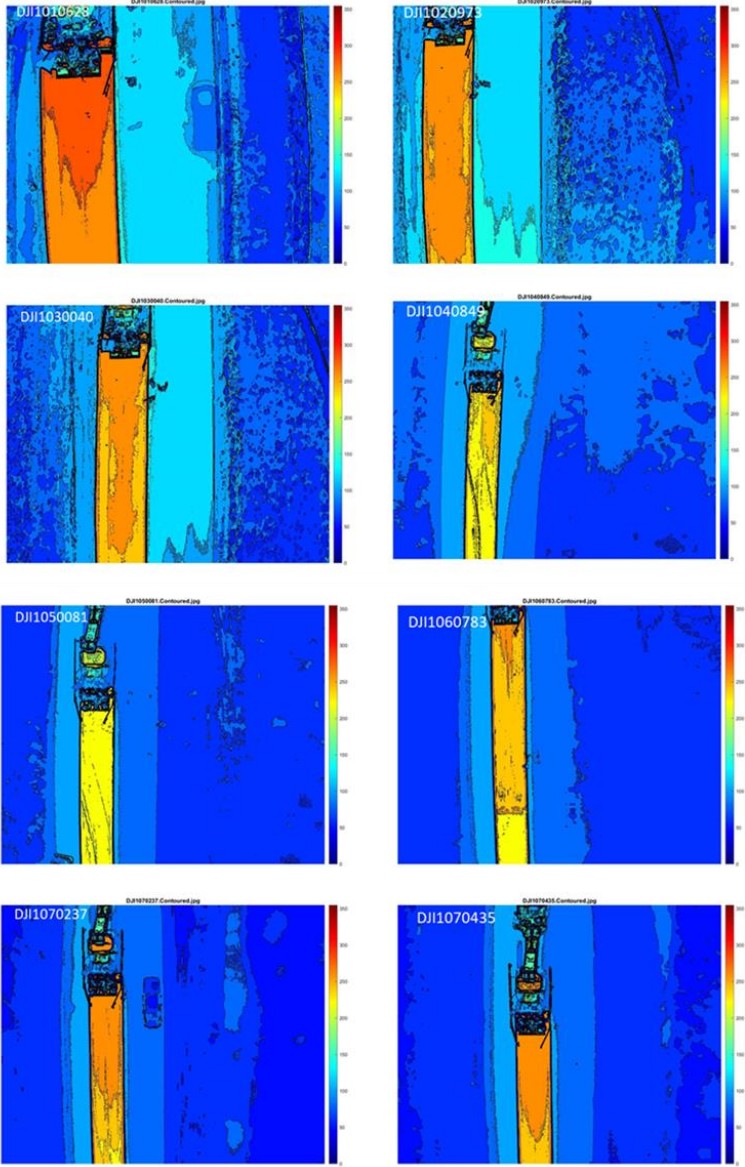

**Figure 12.** Selected contoured images.

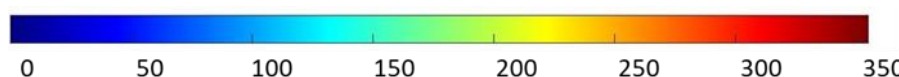

**Figure 13.** Color bar for contoured images.

### 4.3. Data Analysis and Results

According to the Inspector's Daily Report (IDR): "The engineering technician measured temperature of load by using an infrared thermometer and reported Behind paver mix temperature average between 277 °F (136 °C) and 310 °F (154 °C)." The average mat temperature for these eight images is 242 °F (117 °C). This shows a great difference between the manual inspection results, where limited samples are taken, and our proposed methodology to create a temperature profile of the mat. To verify that, we measured the mat temperature immediately behind the paver, using a handheld thermometer. According to that, the mat temperature was between 258 °F (126 °C) and 290 °F (143 °C) immediately behind the paver screed, with an average temperature of 277 °F (136 °C).

After processing the images, they can be studied closely to estimate the thermal parameters and compare them with the manual inspection results. Figure 14 presents the

images with thermal values within the range of 185 °F (85 °C) to 293 °F (145 °C) as an acceptable temperature range (parameter 1 in Table 1). This parameter is visualized on each image, aiming to assist in estimating the value of the first parameter, $R_c$. Equation (2) is used to estimate $R_c$ for each selected image, and the results are summarized in Table 5. The next step is to find the temperature differentials and to estimate the second and third parameters, $R_{ps}$ and $R_{hs}$, as in Equations (3) and (4), respectively.

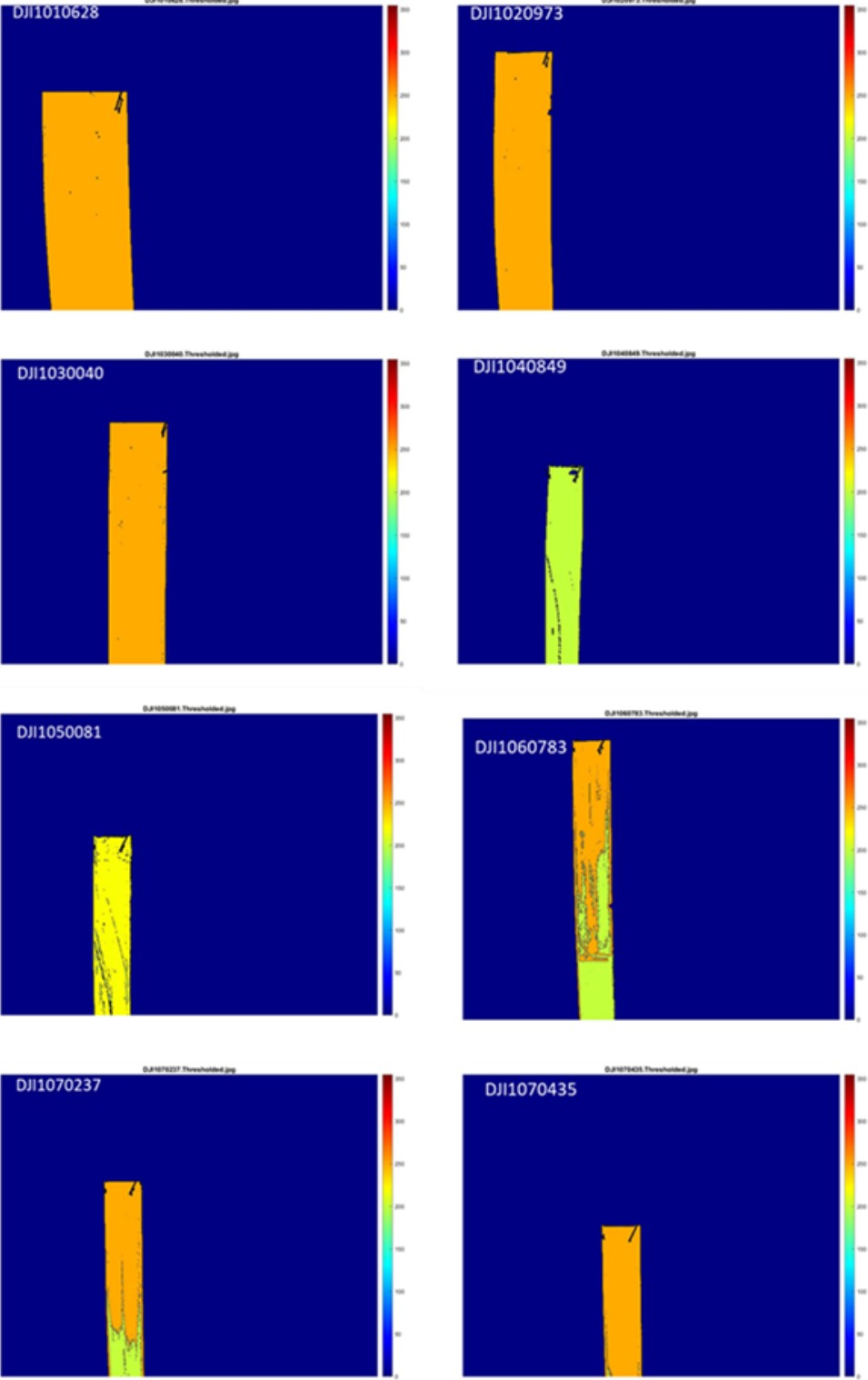

**Figure 14.** Thresholded images within 185 °F (85 °C) to 293 °F (145 °C).

**Table 5.** Estimated segregation parameters for selected images.

| Image | %$R_c$ | %$R_{ps}$ | %$R_{hs}$ |
|---|---|---|---|
| DJI1010628 | 5.3 | 0.4 | 0.8 |
| DJI1020973 | 1.7 | 0.4 | 0.8 |
| DJI1030040 | 0.2 | 0.2 | 0.5 |
| DJI1040849 | 1.4 | 1.8 | 2 |
| DJI1050081 | 1.1 | 1.5 | 1.3 |
| DJI1060783 | 0.7 | 1.2 | 1 |
| DJI1070237 | 0.4 | 0.7 | 1.1 |
| DJI1070435 | 0.7 | 0.4 | 1.6 |

The first image (DJI1010628) has the highest cool area ratio of 5.3% (Figure 15). This image can be analyzed in detail to find the most problematic locations. For this purpose, $R_c$ is calculated for all cross-sections across the road in Image DJI1010628. To visually represent the results, a contour is applied showing the most problematic cross-sections in "red" and the least problematic in "blue". Figure 16 illustrates these cross-sections across the road. Cross-sections shown in red on the image are the most critical ones, with $R_c$ about 13%.

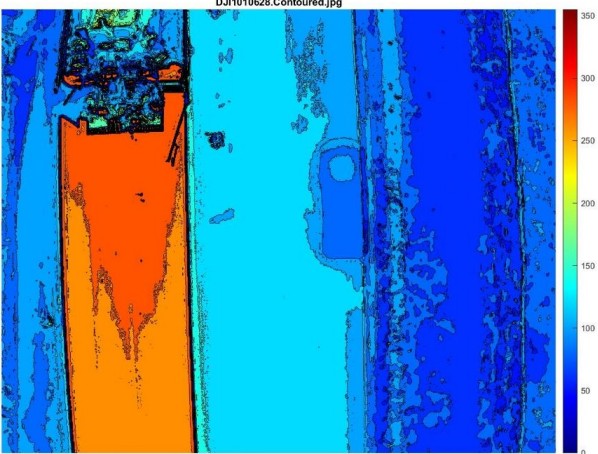

**Figure 15.** Image DJI1010628 with highest cool area ratio.

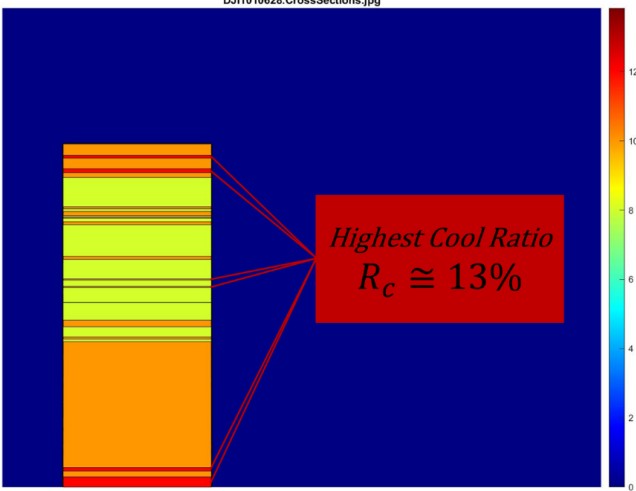

**Figure 16.** Cross-section analysis on Image DJI1010628.

In the presence of a threshold for any of these metrics, MDOT could use these numbers to accept/reject the work. For example, if the MDOT specification indicates a 5% limit for ratio of cool areas ($R_c$), Image DJI1010628 falls into the unacceptable range, and the contractor will be asked to repeat the work (stations associated with flight DJI101) to satisfy the temperature requirements of the mat.

## 5. Discussion

This paper presented a proposed workflow that utilizes a UAS for thermal segregation inspection in HMA and automates the data analysis process through image processing. The methodology was specifically tested on an MDOT project involving three key steps: (a) data collection, (b) data processing, and (c) data analysis and result interpretation. The collected data successfully demonstrated the capability to detect temperature ranges, temperature differentials, and segregated areas on the HMA mat.

The main contribution of this research is directed at agencies in the infrastructure domain, such as DOTs. This research contributes to the body of knowledge by creating a formal workflow for Automated Pavement Construction Inspection using a UAS.

The implementation of the proposed methodology is subject to various limitations that should be taken into account. These limitations include factors such as Federal Aviation Administration (FAA) flying permission constraints, the availability of trained personnel, expertise in data processing techniques, and the potential influence of weather conditions. The FAA imposes certain conditions and limitations, such as restricting the maximum weight of the UAS to 55 lbs (25 kg), requiring a pilot in command and a visual observer, and mandating that the UAS remains within the visual line of sight. These limitations are comprehensively explained by the authors in [24].

Lastly, this study was conducted as part of the phase-three research project titled "Integration of Unmanned Aerial Systems Data Collection into Day-to-Day Usage for Transportation Infrastructure" [24]. In the second phase, the project focused on analyzing the cost and benefits of utilizing a UAS for bridge inspections [18]. However, the scope of the third phase of the research project did not include a similar analysis for the application of a UAS in HMA thermal inspection. It is important to acknowledge that the workflow proposed in this paper is intended to support labor and streamline the workforce. However, conducting a direct comparison between traditional and automated workflows may not be feasible at present. This is primarily because the cost factor relies on the interplay of supply and demand, and the implementation of the proposed UAS technology lacks information regarding supply and demand dynamics necessary for a comprehensive cost analysis.

## 6. Future Research

DOTs, project managers and inspectors are the potential recipients of the thermal segregation analysis results. The hypothesis is that by equipping the project managers with segregated areas (stations along the road) on the HMA mat, they will be able to associate it with potential premature failure and repair (patching) areas in the asset management phase. This is expected to ultimately assist them in creating feedback to the design team and improve the quality of HMA construction process.

To evaluate this hypothesis, the next step in this research is to monitor two control sections over time and then find the correlation between the inspected segregated areas and premature failure. This will assist authors in evaluating the effectiveness of the developed automated inspection methodology in improving pavement performance.

In addition, there are several areas that were not within the scope of this paper. These areas can be studied in future to further the developed workflow. The key recommendation is density profiling of cool areas. Locations of potentially segregated areas (or cool areas in general) can be used to test the mat density after compaction. A density profile of the target locations can be developed, using several techniques such as Core Method, Nuclear Density Gauge, or Non-Nuclear Density Method. This will support the methodology

developed in this paper by studying the correlation between temperature differentials and density differentials.

In addition, the workflow illustrated in this paper can be fully automated and implemented on site, and in a near real-time manner, in the presence of appropriate technical hardware. This equips the project managers and inspectors with reliable methods to inspect the mat temperature and take immediate action to correct the causes of thermal segregation. It will reduce the need to revise the work and the need for future repairs.

Lastly, the effectiveness of the proposed methodology will be examined through the analysis of multiple projects. This evaluation aims to unveil the contributions of the proposed automated workflow in terms of usability, accuracy, and interoperability.

**Author Contributions:** The authors confirm contributions to the paper as follows: study conception and design: R.S., A.M. and C.N.B.; data collection: R.S., A.M. and C.N.B.; analysis and interpretation of results: R.S. and A.M.; draft manuscript preparation: R.S. All authors have read and agreed to the published version of the manuscript.

**Funding:** The work presented in this paper is a part of a project funded by MDOT, research project number OR19-064, contract and authorization numbers 2019-0311/Z1, titled "Integration of Unmanned Aerial Systems Data Collection into Day-to-Day Usage for Transportation Infrastructure".

**Data Availability Statement:** Data results have been shared with the Michigan Department of Transportation (MDOT) and can be available on request through the Corresponding Author with approval of MDOT.

**Acknowledgments:** All support from MDOT and MTRI is gratefully acknowledged, including Michael Meyer, Steve Cook, and Andre Clover at MDOT for project advice, and Rick Dobson, Chris Cook, Abby Jenkins, Vanessa Barber, and Julie Carter at MTRI for UAS data collection. Any opinions, findings, conclusions, or recommendations presented in this paper are those of the authors and do not necessarily reflect the views of the funding agencies.

**Conflicts of Interest:** The authors declare no conflict of interest.

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
