# Peer review of "Automated Pavement Construction Inspection Using Uncrewed Aerial Systems (UAS)—Hot Mixed Asphalt (HMA) Temperature Segregation"

_drones, doi:10.3390/drones7070419_

Round 1

Reviewer 1 Report

1. The paper addresses an important issue in HMA pavement construction, namely temperature segregation, which can lead to performance problems and reduced fatigue life. This is a significant concern for the transportation industry.

2. The abstract clearly highlights the limitations of traditional inspection processes and the need for an alternative solution that is remote, non-destructive, safe, and efficient. This indicates a practical motivation for the research.

3. The objective of the research, as stated in the abstract, is to investigate an automated UAS imaging framework for HMA pavement construction inspection, with a focus on locating temperature segregation. This objective provides a clear direction for the study.

4. The primary contribution of the research, mentioned in the abstract, is to provide workflows for creating enhanced remote inspection procedures and detailed thermal profiles of the placed HMA mat. This contribution has practical implications for departments of transportation and contractors.

5. It would be beneficial for the paper to provide a comprehensive overview of the proposed UAS imaging framework, including details on the specific UAS technology utilized, the imaging techniques employed, and the data analysis methods implemented.

6. The abstract mentions a case study conducted in Michigan to illustrate the application of the proposed framework. It would be valuable to provide more information on the scale and scope of the project, as well as any significant findings or results obtained.

7. To support the claims made regarding the superiority of the proposed framework, the paper could include tables comparing the performance of the UAS imaging approach with traditional manual inspection processes. These tables could demonstrate metrics such as inspection time, accuracy in locating temperature segregation, and safety improvements.

8. The abstract suggests that the proposed framework is remote, non-destructive, safe, and efficient. It would be helpful for the paper to provide more specific details on how the framework achieves these characteristics and what makes it superior to existing methods.

9. The abstract could mention the potential cost savings associated with the implementation of the proposed framework. If the UAS imaging approach proves to be more efficient and accurate than manual inspection, it could lead to reduced costs in terms of labor, equipment, and material waste.

10. It would be valuable for the paper to discuss any potential limitations or challenges encountered during the implementation of the UAS imaging framework. This would provide a more balanced view of the proposed solution and help readers understand its practical implications in real-world scenarios.

Language and Structure: The abstract is written in a clear and concise manner, effectively conveying the purpose, objectives, and potential impact of the research. The flow of information is logical and allows the reader to grasp the key points easily.

Author Response

Reviewer 1:

  1. The paper addresses an important issue in HMA pavement construction, namely temperature segregation, which can lead to performance problems and reduced fatigue life. This is a significant concern for the transportation industry.

Thank you for your thoughtful and encouraging comment. Your positive feedback is highly appreciated. We are grateful for your support and for recognizing the value of our work.

  1. The abstract clearly highlights the limitations of traditional inspection processes and the need for an alternative solution that is remote, non-destructive, safe, and efficient. This indicates a practical motivation for the research.

Thank you for acknowledging this.

  1. The objective of the research, as stated in the abstract, is to investigate an automated UAS imaging framework for HMA pavement construction inspection, with a focus on locating temperature segregation. This objective provides a clear direction for the study.

Thank you for acknowledging this.

  1. The primary contribution of the research mentioned in the abstract is to provide workflows for creating enhanced remote inspection procedures and detailed thermal profiles of the placed HMA mat. This contribution has practical implications for departments of transportation and contractors.

Thank you for acknowledging this.

  1. It would be beneficial for the paper to provide a comprehensive overview of the proposed UAS imaging framework, including details on the specific UAS technology utilized, the imaging techniques employed, and the data analysis methods implemented.

The data collection section is extended to include more detailed information and address this comment in Lines 213-242 as follows:

“Figure 3 captures the thermal data collection by MTRI at I-69 project. Before flying, the process starts with placing ground control points (GCPs) (Figure 3, A) to provide high accuracy georeferencing information for the collected imagery.  The GCPs used here were Aeropoint units from Propeller, which provided approximately 3cm positional accuracy for the center of each target using built-in GPS and post-processing capabilities. This positional information improves the output quality of registered and merged images.

For the imaging part of the UAS framework, missions were pre-planned for the areas of interest using the DJI Pilot application that is available on the DJI Smart Controller. The Pilot app includes the ability to set missions that are focused on getting a dense network of overlapping thermal images, with 80% front and side overlap, in case a thermal orthophoto is needed. Heights can vary, but typically 24 meters (80 feet) was useful for thermal missions. Optical (natural color) and thermal images were taken at the maximum rate of two frames per second. The boundaries of the mission area were drawn on the interactive Pilot app screen, with the thermal mission option, with normal flying speeds around 4.2 kph (2.6 mph) with the 24 m flying height. Caution was taken to make sure that the UAS would not be flying over any moving traffic. The pilot in command (PIC) was ready to take control from the automated mission at all times.

Weather conditions were checked ahead of each flight starting 48 hours ahead of time using U.S. National Weather Service and smart phone apps such as Windy and UAV Forcast, and then the day of the data collection, including up to right before take-off. Generally, the PIC would plan for collecting in steady wind speeds of no more than 24 kph (15 mph) with wind gusts below 40 kph (25 mph).”

  1. The abstract mentions a case study conducted in Michigan to illustrate the application of the proposed framework. It would be valuable to provide more information on the scale and scope of the project, as well as any significant findings or results obtained.

This comment is addressed in Lines 305-309 as follows:

“To illustrate the proposed workflow, Michigan DOT’s reconstruction project of I-69 located in South Lansing, Ainger Road was studied. Table 3 provides the information for this project for the I-69 MDOT project case study area. More information about this project has been posted at [31 and 32]. The project has focused on extending the life of the existing infrastructure.”

  1. To support the claims made regarding the superiority of the proposed framework, the paper could include tables comparing the performance of the UAS imaging approach with traditional manual inspection processes. These tables could demonstrate metrics such as inspection time, accuracy in locating temperature segregation, and safety improvements.

This is a great recommendation. While the previous phase of this research has investigated these improvements using the UAV, this study did not specifically consider a similar comparison. To address this comment, Lines 414-424 are added:

“Lastly, this study was conducted as part of the phase three research project titled "Integration of Unmanned Aerial Systems Data Collection into Day-to-Day Usage for Transportation Infrastructure" [24]. In the second phase, the project focused on analyzing the cost and benefits of utilizing UAS for bridge inspections [18]. However, the scope of the third phase research did not include a similar analysis for the application of UAS in HMA thermal inspection. It is important to acknowledge that the workflow proposed in this paper is intended to support labor and streamline the workforce. However, conducting a direct comparison between traditional and automated workflows may not be feasible at present. This is primarily because the cost factor relies on the interplay of supply and demand, and the implementation of the proposed UAS technology lacks information regarding supply and demand dynamics necessary for a comprehensive cost analysis.”

  1. The abstract suggests that the proposed framework is remote, non-destructive, safe, and efficient. It would be helpful for the paper to provide more specific details on how the framework achieves these characteristics and what makes it superior to existing methods.

Please refer to comment 7. A future study is designed to enhance the current research, by measuring the efficacy of the proposed methodology.

  1. The abstract could mention the potential cost savings associated with the implementation of the proposed framework. If the UAS imaging approach proves to be more efficient and accurate than manual inspection, it could lead to reduced costs in terms of labor, equipment, and material waste.

The term “efficient” in Line 20 is used to include efficiency in time and cost. However, since the authors have not measured this in terms of cost, no qualitative data is available yet. As mentioned in comment 7, a future study is designed to measure the efficacy of this methodology.

  1. It would be valuable for the paper to discuss any potential limitations or challenges encountered during the implementation of the UAS imaging framework. This would provide a more balanced view of the proposed solution and help readers understand its practical implications in real-world scenarios.

This comment is addressed in Lines 406-413 as follows:

“The implementation of the proposed methodology is subject to various limitations that should be taken into account. These limitations include factors such as Federal Aviation Administration (FAA) flying permission constraints, availability of trained personnel, expertise in data processing techniques, and potential influence of weather conditions. The FAA imposes certain conditions and limitations, such as restricting the maximum weight of the UAS to 55 lbs (25 kg), requiring a pilot in command and a visual observer, and mandating that the UAS remains within the visual line of sight. These limitations are comprehensively explained by the authors in [24].”

Reviewer 2 Report

Automated Pavement Construction Inspection Using Uncrewed 2 Aerial Systems (UAS) – HMA Temperature Segregation. After careful reading I found the following:

1- The references used in the document are not not enough. It is important to have a sufficient number of references in any document in order to support the ideas and arguments presented.

2- The methodolgy is not clear, and the findings are not well-supported.

3- The disscution part is missing, and there is no clear conclusion or summary of the findings.

4- Some figure citations is missing, see line 310.

5- This work is sutable to be a report than a research paper, as it lacks depth and complexity.

Author Response

Reviewer 2:

Automated Pavement Construction Inspection Using Uncrewed 2 Aerial Systems (UAS) – HMA Temperature Segregation. After careful reading I found the following:

Thank you for taking the time to review our work. We appreciate your input. However, we are only able to respond to specific and detailed comments to better identify the areas of improvement. All comments are addressed by the reviewers and the paper is revised to reflect them.

  1. The references used in the document are not enough. It is important to have a sufficient number of references in any document in order to support the ideas and arguments presented.

As mentioned in the paper, there are only a few papers that have studied application of UAS on HMA thermal segregation inspection. We have extended our references to include more publications.

  1. The methodology is not clear, and the findings are not well-supported.

Thank you for providing your feedback on our methodology and findings. We value your perspective, though we respectfully disagree with this comment. We have made efforts to clearly outline our methodology, including the steps involved in data collection, data processing, and data analysis. Additionally, we have presented our findings based on the analysis of the collected data, supporting them with relevant example projects.

  1. The discussion part is missing, and there is no clear conclusion or summary of the findings.

A discussion section is added to address this comment, in Lines 396-424 as follows:

“This paper presented a proposed workflow that utilizes UAS for thermal segregation inspection in HMA and auto-mates the data analysis process through Image Processing. The methodology was specifically tested on an MDOT project, involving three key steps: (a) data collection, (b) data processing, and (c) data analysis and result interpretation. The collected data successfully demonstrated the capability to detect temperature ranges, temperature differentials, and segregated areas on the HMA mat.

The main contribution of this research is directed at agencies in the infrastructure domain such as DOTs. This research contributes to the body of knowledge by creating a formal workflow for Automated Pavement Construction Inspection using UAS.

The implementation of the proposed methodology is subject to various limitations that should be taken into account. These limitations include factors such as Federal Aviation Administration (FAA) flying permission constraints, availability of trained personnel, expertise in data processing techniques, and potential influence of weather conditions. The FAA imposes certain conditions and limitations, such as restricting the maximum weight of the UAS to 55 lbs (25 kg), requiring a pilot in command and a visual observer, and mandating that the UAS remains within the visual line of sight. These limitations are comprehensively explained by the authors in [24].

Lastly, this study was conducted as part of the phase three research project titled "Integration of Unmanned Aerial Systems Data Collection into Day-to-Day Usage for Transportation Infrastructure" [24]. In the second phase, the project focused on analyzing the cost and benefits of utilizing UAS for bridge inspections [18]. However, the scope of the third phase research did not include a similar analysis for the application of UAS in HMA thermal inspection. It is important to acknowledge that the workflow proposed in this paper is intended to support labor and streamline the workforce. However, conducting a direct comparison between traditional and automated workflows may not be feasible at present. This is primarily because the cost factor relies on the interplay of supply and demand, and the implementation of the proposed UAS technology lacks information regarding supply and demand dynamics necessary for a comprehensive cost analysis.”

  1. Some figure citation is missing, see line 310.

We have revised the cross-reference for “Figure 14”. It looks like an internal error on Word Program, since the reference was initially visible on our system.

  1. This work is suitable to be a report than a research paper, as it lacks depth and complexity.

Thank you for your feedback on this paper. While we appreciate your perspective on the depth and complexity of the paper, we respectfully disagree with the assessment that it is more suitable as a report rather than a research paper. We believe that our work contributes to the existing body of knowledge in automated thermal inspection using UAS and offers valuable insights. We have endeavored to provide a comprehensive analysis and present our findings in a clear and organized structure. A detailed review does a better job in helping us enhance the quality of our paper.

Reviewer 3 Report

This is an interesting, well-written article investigating a UAS-based imaging framework for HMA pavement construction inspection. It also reports the results of a case study where the proposed framework is implemented and compared to the traditional inspection method. The manuscript could also benefit from further improvements, as follows:

Minor:

Spell out the acronym ‘HMA’ in the abstract and the body text

Line 107: the word “Images” should not have an uppercase I

Line 178: “illustrates” (present tense)

Line 180: “identifies” (present tense)

Line 258: The readers should not be required to know where Michigan is located. Please identify the country.

Major:

Line 105: the statement “there is limited research on using UAS-mounted thermal cameras for thermal profiling purposes” sounds inaccurate. There are many publications on the topic, some describing cutting-edge solutions combining UAS, thermal imaging, and computer vision algorithms that can detect subtle colour/temperature variations in thermal images. I recommend providing evidence of “limited research” or rephrasing.

The above comment also relates to a general concern about this article: 18 references seem too few. There are certainly many more publications on UAS-based thermal imaging in the construction/built environment domain. I recommend expanding the references list to clarify the article’s contribution, contrasting its findings against previous research.

Most of the “literature Review” subsection “Current State of Practice – MDOT” sounds like a summary of the HMA production manual. While it is important to provide information on the traditional inspection process to contextualize and justify the proposed innovation, this section can be further summarized. Line 161 onwards provides the most relevant info.

Is the proposed framework also the research methodology? This is a little confusing. The methodology section of an academic article should describe the research methodology, that is, the research approach and set of methods utilized to achieve the research objectives (in your case, the framework itself). Instead, the current “methodology” section is actually describing the automated inspection framework developed, which is part of the research results/outputs, not the means to produce it. There is no description of how the framework was developed. Was it through trial and error? Intuitively? Based on any previous studies?

In section 4, the case study description, there is something about comparing the drone-based inspection results with the manual inspection results (using other instruments such as thermometers). Assuming this comparison is part of the research objectives (otherwise, why conduct it?), it should be mentioned in the abstract and introduction/objectives sections and described in detail in the methodology section.

In section 5, findings should be linked to previous research - how does this study add to the body of knowledge? Also, other major limitations are not included.

Finally, I am not convinced whether the proposed framework provides an “automated” inspection process. The fact that it uses drone thermal images does not make it automated. There is still a high degree of manual labour – data collection (by manually flying the drone) and processing (very labour-intensive, random sampling of individual images from datasets, a lot of space for human error). In my view, the proposed framework should not be called an “automated” one, and this should be corrected in the title and throughout the manuscript.

Author Response

Reviewer 3:

This is an interesting, well-written article investigating a UAS-based imaging framework for HMA pavement construction inspection. It also reports the results of a case study where the proposed framework is implemented and compared to the traditional inspection method. The manuscript could also benefit from further improvements, as follows:

We appreciate your recognition of the interesting and well-written nature of the paper. Your suggestions for further improvements are carefully considered as we continue to enhance the manuscript.

Minor:

  1. Spell out the acronym ‘HMA’ in the abstract and the body text.

To address this comment, HMA is spelled out in both abstract and introduction sections.

  1. Line 107: the word “Images” should not have an uppercase I

To address this problem, “Images” is changed to “images”.

  1. Line 178: “illustrates” (present tense)

To address this problem, “illustrated” is changed to “illustrates”.

  1. Line 180: “identifies” (present tense)

To address this problem, “identified” is changed to “identifies”.

  1. Line 258: The readers should not be required to know where Michigan is located. Please identify the country.

To address this comment, “US” is added to Line 81 as follows:

“This section also reviews the current state of practice for HMA manual inspection in Michigan, US.”

Major:

  1. Line 105: the statement “there is limited research on using UAS-mounted thermal cameras for thermal profiling purposes” sounds inaccurate. There are many publications on the topic, some describing cutting-edge solutions combining UAS, thermal imaging, and computer vision algorithms that can detect subtle colour/temperature variations in thermal images. I recommend providing evidence of “limited research” or rephrasing.

While UAS has been used for other automated construction inspection purposes, its application in “HMA Thermal Segregation Inspection” has been limited, in comparison to other applications and tools. This is revisited in Lines 122-131 as following:

“While several recent studies [17, 18, 19, 20, 21, 22, 23, 24, 25, 26, 27, and 28] have investigated the application of UAS on pavement distress detection, there is limited research on utilizing UAS for HMA thermal segregation inspection. For instance, Du et al. [23] reviewed the application of digital Image Processing tools on pavement distress detection, including thermal segregation. In their study, they have reviewed several data acquisition methods such as (a) cameras, (b) thermal imaging technologies, (c) laser, and (d) ground generation radar. Those authors have also made a comparison between different imaging tools and reviewed their advantages and limitations. Some of the benefits of using thermal imaging technologies are stated as convenient portability, contactless testing, absence of harmful radiation, and the ability to capture real-time images.

In addition, there is no automated workflow proposed for HMA construction inspection, using thermal images. To address these two gaps, applications, and limitations of using UAS in an automated construction inspection workflow needs to be studied…”

  1. The above comment also relates to a general concern about this article: 18 references seem too few. There are certainly many more publications on UAS-based thermal imaging in the construction/built environment domain. I recommend expanding the references list to clarify the article’s contribution, contrasting its findings against previous research.

Please refer to the previous comment regarding the updates. We have updated our references to add more depth, however there is limited research on this particular application.

  1. Most of the “literature Review” subsection “Current State of Practice – MDOT” sounds like a summary of the HMA production manual. While it is important to provide information on the traditional inspection process to contextualize and justify the proposed innovation, this section can be further summarized. Line 161 onwards provides the most relevant info.

We appreciate your suggestion. We have assumed that the readers might not be familiar with the MDOT’s existing workflows, and the provided information is to introduce them to the traditional approach to inspecting HMA paving activity.

  1. Is the proposed framework also the research methodology? This is a little confusing. The methodology section of an academic article should describe the research methodology, that is, the research approach and set of methods utilized to achieve the research objectives (in your case, the framework itself). Instead, the current “methodology” section is actually describing the automated inspection framework developed, which is part of the research results/outputs, not the means to produce it. There is no description of how the framework was developed. Was it through trial and error? Intuitively? Based on any previous studies?

To address this misunderstanding, we have revised the document in Lines 202-204 as follows:

“In this section, an automated inspection methodology is proposed and explained in three subsections of (a) data collection, (b) data processing, and (c) data analysis and result.”

As mentioned in the funding section, this project was a part of a bigger project, titled “Integration of Unmanned Aerial Systems Data Collection into Day-to-Day Usage for Transportation Infrastructure”. The proposed methodology builds upon the experience authors have gained through three phases of this research.

  1. In section 4, the case study description, there is something about comparing the drone-based inspection results with the manual inspection results (using other instruments such as thermometers). Assuming this comparison is part of the research objectives (otherwise, why conduct it?), it should be mentioned in the abstract and introduction/objectives sections and described in detail in the methodology section.

Thank you for the insightful recommendation. While the previous phase of this research has investigated these improvements using the UAV, this study did not specifically consider a similar comparison. To address this comment, Lines 414-424 are added:

“Lastly, this study was conducted as part of the phase three research project titled "Integration of Unmanned Aerial Systems Data Collection into Day-to-Day Usage for Transportation Infrastructure" [24]. In the second phase, the project focused on analyzing the cost and benefits of utilizing UAS for bridge inspections [18]. However, the scope of the third phase research did not include a similar analysis for the application of UAS in HMA thermal inspection. It is important to acknowledge that the workflow proposed in this paper is intended to support labor and streamline the workforce. However, conducting a direct comparison between traditional and automated workflows may not be feasible at present. This is primarily because the cost factor relies on the interplay of supply and demand, and the implementation of the proposed UAS technology lacks information regarding supply and demand dynamics necessary for a comprehensive cost analysis.”

  1. In section 5, findings should be linked to previous research - how does this study add to the body of knowledge? Also, other major limitations are not included.

We have created subsections under the case study section, to better explain the flow of the project. We have also added a discussion section. Limitations are discussed in Lines 406-413 as follows:

“The implementation of the proposed methodology is subject to various limitations that should be taken into account. These limitations include factors such as Federal Aviation Administration (FAA) flying permission constraints, availability of trained personnel, expertise in data processing techniques, and potential influence of weather conditions. The FAA imposes certain conditions and limitations, such as restricting the maximum weight of the UAS to 55 lbs (25 kg), requiring a pilot in command and a visual observer, and mandating that the UAS remains within the visual line of sight. These limitations are comprehensively explained by the authors in [24].”

  1. Finally, I am not convinced whether the proposed framework provides an “automated” inspection process. The fact that it uses drone thermal images does not make it automated. There is still a high degree of manual labour – data collection (by manually flying the drone) and processing (very labour-intensive, random sampling of individual images from datasets, a lot of space for human error). In my view, the proposed framework should not be called an “automated” one, and this should be corrected in the title and throughout the manuscript.

Thank you for raising the point about the level of automation in our paper. While the current implementation involves manual labor in certain aspects such as data collection and processing, we acknowledge the potential for further advancements towards achieving full automation.

Reviewer 4 Report

This paper presents an automated pavement inspection workflow using Uncrewed Aerial Systems (UAS) and image processing techniques. The objective of the paper is clear, and the paper is well-organized. This review suggests several comments to improve the quality of the manuscript.

ABSTRACT, BACKGROUND, and LITERATURE 

Although the term “HMA, Hot mix asphalt” is common in this field, the full name of the abbreviation should be presented for readers who may not be in the field.

In lines 66 – 68, the authors mentioned that the developed framework does not depend on UAS as the only data collection tool. It can be expanded to incorporate any other data collection tool, such as handheld IR cameras. However, this statement seems to weaken the significant assumption made by the authors; UAS was utilized for the safety of the inspector and efficient inspection without a site visit. The handheld IR cameras also require a site visit. Moreover, the later part of the paper does not present any supporting information and evidence for this statement.

Figure 1 may need a citation unless it was created by the authors.

The literature review can be improved because the current review focuses on the standards and criteria. However, because this paper is focused on the detection of temperature using UAS, the literature on temperature detection methods and UAS applications should be added.

METHODOLOGY AND RESULT

From the scaled infrared images, the authors counted the number of pixels to calculate three parameters. This approach will work with images that have clearly distinguishable temperatures. However, what if the boundary is not clear enough? For example, in Figure 4 B, the asphalt pavement and adjacent roads (existing pavement based on the authors’ expression) are both yellow (orange) color. If the authors classify the new pavement based on the intensity of IR images. What is the threshold value to classify them?

In line 310, there is a citation error.

DISCUSSION and CONCLUSION

To validate the effectiveness of the proposed framework, this reviewer suggests a comparison of the proposed framework with the traditional inspection method. This may include the evaluation of the temperature measurement accuracy of the IR cameras mounted on the drone (or relevant literature) and a comparison of the inspection results from both approaches at the same time.

Author Response

Reviewer 4:

This paper presents an automated pavement inspection workflow using Uncrewed Aerial Systems (UAS) and image processing techniques. The objective of the paper is clear, and the paper is well-organized. This review suggests several comments to improve the quality of the manuscript.

We appreciate your recognition of the clear objective and the overall organization of the paper. We have considered all your comments in order to improve the quality of the manuscript.

ABSTRACT, BACKGROUND, and LITERATURE

  1. Although the term “HMA, Hot mix asphalt” is common in this field, the full name of the abbreviation should be presented for readers who may not be in the field.

We have addressed this comment by adding the full name, the first time HMA appears in the abstract, and introduction.

  1. In lines 66 – 68, the authors mentioned that the developed framework does not depend on UAS as the only data collection tool. It can be expanded to incorporate any other data collection tool, such as handheld IR cameras. However, this statement seems to weaken the significant assumption made by the authors; UAS was utilized for the safety of the inspector and efficient inspection without a site visit. The handheld IR cameras also require a site visit. Moreover, the later part of the paper does not present any supporting information and evidence for this statement.

This statement seems to be misleading. We have removed it from the document to make the research scope clear.

  1. Figure 1 may need a citation unless it was created by the authors.

This comment is address by adding the reference to Figure 1 in Line 159 as:

“Figure 1 Sampling Pattern Behind the Paver – Picture is taken from [7]”

  1. The literature review can be improved because the current review focuses on the standards and criteria. However, because this paper is focused on the detection of temperature using UAS, literature on temperature detection methods and UAS applications should be added.

While most of the previous work in focused on “progress monitoring” and “bridge inspection” applications, there are only a few applications in using UAS for HMA thermal segregation inspection. This is included in Line 122-131 as follows:

“While several recent studies [17, 18, 19, 20, 21, 22, 23, 24, 25, 26, 27, and 28] have investigated the application of UAS on pavement distress detection, there is limited research on utilizing UAS for HMA thermal segregation inspection. For instance, Du et al. [23] reviewed the application of digital Image Processing tools on pavement distress detection, including thermal segregation. In their study, they have reviewed several data acquisition methods such as (a) cameras, (b) thermal imaging technologies, (c) laser, and (d) ground generation radar. Those authors have also made a comparison between different imaging tools and reviewed their advantages and limitations. Some of the benefits of using thermal imaging technologies are stated as convenient portability, contactless testing, absence of harmful radiation, and the ability to capture real-time images.”

METHODOLOGY AND RESULT

  1. From the scaled infrared images, the authors counted the number of pixels to calculate three parameters. This approach will work with images that have clearly distinguishable temperatures. However, what if the boundary is not clear enough? For example, in Figure 4 B, the asphalt pavement and adjacent roads (existing pavement based on the authors’ expression) are both yellow (orange) color. If the authors classify the new pavement based on the intensity of IR images. What is the threshold value to classify them?

Figure 4.B is an example of images we took after the paving operation was finished. As you mentioned, the difference between the already paved road and the unpaved one is not clearly distinguishable. However, the selected images are from ongoing paving activity, where there is clearly a difference between the sections being paved, and the sections that are not paved. Refer to Figure 5 and any other figure after it, where the edges are better detectable (yellow and blue/purple colors). Lines 346-347 explains the ranges as follows:

“As illustrated in Figure 12, the mat temperature is generally in the range of (200°F, 300°F). while the daylighting and the unpaved shoulder are in the range of (20°F, 100°F).”

  1. In line 310, there is a citation error.

We have revised the cross-reference for “Figure 14”. It looks like an internal error on Word Program, since the reference was initially visible on our system.

DISCUSSION and CONCLUSION

  1. To validate the effectiveness of the proposed framework, this reviewer suggests a comparison of the proposed framework with the traditional inspection method. This may include the evaluation of the temperature measurement accuracy of the IR cameras mounted on the drone (or relevant literature) and a comparison of the inspection results from both approaches at the same time.

Thank you for this valuable recommendation. While the previous phase of this research has investigated the effectiveness of using UAS for construction inspection, this study did not specifically consider a similar comparison. The authors collected ground truth control data to make a comparison between the data collected using the traditional method, and the automated method. However, the traditional data ended up lacking location (GPS) data and made the comparison impossible.

The authors are working on a study to compare the manual and automated methods, in terms of efficacy. To address this comment, Lines 414-424 are added:

Lastly, this study was conducted as part of the phase three research project titled "Integration of Unmanned Aerial Systems Data Collection into Day-to-Day Usage for Transportation Infrastructure" [24]. In the second phase, the project focused on analyzing the cost and benefits of utilizing UAS for bridge inspections [18]. However, the scope of the third phase research did not include a similar analysis for the application of UAS in HMA thermal inspection. It is important to acknowledge that the workflow proposed in this paper is intended to support labor and streamline the workforce. However, conducting a direct comparison between traditional and automated workflows may not be feasible at present. This is primarily because the cost factor relies on the interplay of supply and demand, and the implementation of the proposed UAS technology lacks information regarding supply and demand dynamics necessary for a comprehensive cost analysis.”

Reviewer 5 Report

Thank you for providing me the opportunity to read the article “Automated Pavement Construction Inspection Using Uncrewed Aerial Systems (UAS) – HMA Temperature Segregation”. I have the following comments to improve the article.

·         Abstract lines 19-20 “The objective of this research is to investigate an automated UAS imaging framework for HMA pavement construction inspection, mainly locating temperature segregation.” Are you proposing a framework or investigating an existing one? This is not clear currently.

·         In the introduction, before digging into the types of segregation, it is important to present an understandable definition of the term segregation for non-transport-related readers. Given that the scope of the journal drones is not limited to transportation (or road works) it is important to explain all field-specific technical terms for the journal readers.

·         There are many abbreviations in the paper, such as HMA, and AASHTO, that are not abbreviated on first appearance. While HOT MIXED ASPHALT (HMA) may be well known to transport-related readers, the general readers may struggle to understand it unless a full form is presented. Given the bulk of abbreviations in the paper, it is important that the authors add an abbreviation table to the paper, preferably at the start of the paper.

·         While the authors have touched on the use of UAS as a potential tool for addressing HMA inspection issues, it is not focused on how the current study is novel in its approach. Given multiple studies have used UAS or UAVs for road or bridge (cracks/damage etc.) inspections, the novelty must be clarified and strengthened. Some example studies are as follows:

o   https://www.sciencedirect.com/science/article/pii/S2590123023002578

o   https://rosap.ntl.bts.gov/view/dot/66362

o   https://www.mdpi.com/2071-1050/15/3/1866

o   https://www.mdpi.com/2072-4292/12/18/3084

o   https://link.springer.com/chapter/10.1007/978-3-319-92102-0_24

·         Add “how the paper is organized” in a paragraph at the end of the introduction section.

·         It is confusing to see the terms workflow and frameworks used interchangeably. The current study presents a workflow, not a framework; therefore, the authors should revise the terms throughout the paper and stick to “workflow”.

·         Has the UAS height, flying speed, and wind resistance been considered? These are important parameters to be considered that must be clarified or alternatively, the authors should present their point of view to justify why these are not important or relevant to the current study.

·         Line 310, check “error! Reference source not found” and add the correct ref here.

·         It is important that the results are compared to other similar studies in a “discussion” section. The authors need to highlight how the proposed workflow compares to other similar studies. This should indicate the superior performance or comparable results of the current workflow.

The language is fine, needing minor edits

Author Response

Reviewer 5:

Thank you for providing me the opportunity to read the article “Automated Pavement Construction Inspection Using Uncrewed Aerial Systems (UAS) – HMA Temperature Segregation”. I have the following comments to improve the article.

Thank you for taking the time to review our paper. We appreciate your valuable feedback and look forward to considering your comments for improving this paper.

  1. Abstract lines 19-20 “The objective of this research is to investigate an automated UAS imaging framework for HMA pavement construction inspection, mainly locating temperature segregation.” Are you proposing a framework or investigating an existing one? This is not clear currently.

We have proposed (designed) a new methodology. To address this comment, we have revised Line 20 as:

“The objective of this research is to design an automated UAS imaging workflow for HMA pavement construction inspection, mainly locating temperature segregation.”

  1. In the introduction, before digging into the types of segregation, it is important to present an understandable definition of the term segregation for non-transport-related readers. Given that the scope of the journal drones is not limited to transportation (or road works) it is important to explain all field-specific technical terms for the journal readers.

To address this comment, segregation is defined in Lines 32-33 as follows:

“Segregation, defined as “the non-uniform distribution of coarse and fine aggregate components within the asphalt mixture”, is a common problem in Hot Mixed Asphalt (HMA) pavement construction [1].”

  1. There are many abbreviations in the paper, such as HMA, and AASHTO, that are not abbreviated on first appearance. While HOT MIXED ASPHALT (HMA) may be well known to transport-related readers, the general readers may struggle to understand it unless a full form is presented. Given the bulk of abbreviations in the paper, it is important that the authors add an abbreviation table to the paper, preferably at the start of the paper.

To address this comment, abbreviations are revised and in the first appearance, they are written fully. This is done for both HMA and AASHTO.

  1. While the authors have touched on the use of UAS as a potential tool for addressing HMA inspection issues, it is not focused on how the current study is novel in its approach. Given multiple studies have used UAS or UAVs for road or bridge (cracks/damage etc.) inspections, the novelty must be clarified and strengthened. Some example studies are as follows:
    1. https://www.sciencedirect.com/science/article/pii/S2590123023002578
    2. https://rosap.ntl.bts.gov/view/dot/66362
    3. https://www.mdpi.com/2071-1050/15/3/1866
    4. https://www.mdpi.com/2072-4292/12/18/3084
    5. https://link.springer.com/chapter/10.1007/978-3-319-92102-0_24

Thank you for providing a sample list of research on applications of UAS in construction inspection. While most of the previous work in focused on “progress monitoring” and “bridge inspection” applications, there are only a few applications in using UAS for HMA thermal segregation inspection. This is included in Lines 122-131 as follows:

“While several recent studies [17, 18, 19, 20, 21, 22, 23, 24, 25, 26, 27, and 28] have investigated the application of UAS on pavement distress detection, there is limited research on utilizing UAS for HMA thermal segregation inspection. For instance, Du et al. [23] reviewed the application of digital Image Processing tools on pavement distress detection, including thermal segregation. In their study, they have reviewed several data acquisition methods such as (a) cameras, (b) thermal imaging technologies, (c) laser, and (d) ground generation radar. Those authors have also made a comparison between different imaging tools and reviewed their advantages and limitations. Some of the benefits of using thermal imaging technologies are stated as convenient portability, contactless testing, absence of harmful radiation, and the ability to capture real-time images.”

  1. Add “how the paper is organized” in a paragraph at the end of the introduction section.

This comment is addressed in Lines 78-86 as follows:

“This paper is structured in six sections. The introduction section is followed by a literature review section, where the recent research in HMA thermal segregation is studied. This section also reviews the current state of practice for HMA manual inspection in Michigan, US. Section three elaborates the research methodology, in three steps of (a) data collection, (b) data processing, and (c) data analysis and result. The proposed methodology is illustrated in section four, using an example HMA project, by MDOT. The developed workflow is later discussed in section five, followed by a conclusion in the last section. The limitations of the workflow, as well as the direction for future research are also explained in that section.”

  1. It is confusing to see the terms workflow and frameworks used interchangeably. The current study presents a workflow, not a framework; therefore, the authors should revise the terms throughout the paper and stick to “workflow”.

The term “framework” is revised to “workflow” throughout the paper.

  1. Has the UAS height, flying speed, and wind resistance been considered? These are important parameters to be considered that must be clarified or alternatively, the authors should present their point of view to justify why these are not important or relevant to the current study.

This comment is addressed in Lines 213-242 as follows:

“Figure 3 captures the thermal data collection by MTRI at I-69 project. Before flying, the process starts with placing ground control points (GCPs) (Figure 3, A) to provide high accuracy georeferencing information for the collected imagery.  The GCPs used here were Aeropoint units from Propeller, which provided approximately 3cm positional accuracy for the center of each target using built-in GPS and post-processing capabilities. This positional information improves the output quality of registered and merged images.

For the imaging part of the UAS framework, missions were pre-planned for the areas of interest using the DJI Pilot application that is available on the DJI Smart Controller. The Pilot app includes the ability to set missions that are focused on getting a dense network of overlapping thermal images, with 80% front and side overlap, in case a thermal orthophoto is needed. Heights can vary, but typically 24 meters (80 feet) was useful for thermal missions. Optical (natural color) and thermal images were taken at the maximum rate of two frames per second. The boundaries of the mission area were drawn on the interactive Pilot app screen, with the thermal mission option, with normal flying speeds around 4.2 kph (2.6 mph) with the 24 m flying height. Caution was taken to make sure that the UAS would not be flying over any moving traffic. The pilot in command (PIC) was ready to take control from the automated mission at all times.

Weather conditions were checked ahead of each flight starting 48 hours ahead of time using U.S. National Weather Service and smart phone apps such as Windy and UAV Forcast, and then the day of the data collection, including up to right before takeoff. Generally, the PIC would plan for collecting in steady wind speeds of no more than 24 kph (15 mph) with wind gusts below 40 kph (25 mph).”

  1. Line 310, check “error! Reference source not found” and add the correct ref here.

We have revised the cross-reference for “Figure 14”. It looks like an internal error on Word Program, since the reference was initially visible on our system.

  1. It is important that the results are compared to other similar studies in a “discussion” section. The authors need to highlight how the proposed workflow compares to other similar studies. This should indicate the superior performance or comparable results of the current workflow.

A discussion section is added to address this comment, in Lines 396-424 as follows:

“This paper presented a proposed workflow that utilizes UAS for thermal segregation inspection in HMA and auto-mates the data analysis process through Image Processing. The methodology was specifically tested on an MDOT project, involving three key steps: (a) data collection, (b) data processing, and (c) data analysis and result interpretation. The collected data successfully demonstrated the capability to detect temperature ranges, temperature differentials, and segregated areas on the HMA mat.

The main contribution of this research is directed at agencies in the infrastructure domain such as DOTs. This research contributes to the body of knowledge by creating a formal workflow for Automated Pavement Construction Inspection using UAS.

The implementation of the proposed methodology is subject to various limitations that should be taken into account. These limitations include factors such as Federal Aviation Administration (FAA) flying permission constraints, availability of trained personnel, expertise in data processing techniques, and potential influence of weather conditions. The FAA imposes certain conditions and limitations, such as restricting the maximum weight of the UAS to 55 lbs (25 kg), requiring a pilot in command and a visual observer, and mandating that the UAS remains within the visual line of sight. These limitations are comprehensively explained by the authors in [24].

Lastly, this study was conducted as part of the phase three research project titled "Integration of Unmanned Aerial Systems Data Collection into Day-to-Day Usage for Transportation Infrastructure" [24]. In the second phase, the project focused on analyzing the cost and benefits of utilizing UAS for bridge inspections [18]. However, the scope of the third phase research did not include a similar analysis for the application of UAS in HMA thermal inspection. It is important to acknowledge that the workflow proposed in this paper is intended to support labor and streamline the workforce. However, conducting a direct comparison between traditional and automated workflows may not be feasible at present. This is primarily because the cost factor relies on the interplay of supply and demand, and the implementation of the proposed UAS technology lacks information regarding supply and demand dynamics necessary for a comprehensive cost analysis.”

Reviewer 6 Report

 was very engaged by this paper, which effectively communicates its significant contributions to the field, particularly with its focus on technical applications. The framework introduced in this paper brings substantial novelty by proposing the replacement of manual inspection, data acquisition, and data extraction with Unmanned Aerial Systems (UAS). It is impressive how the authors have delved into the development and outcomes of this framework.
However, the discussion seems to emphasize more on the technical aspects, while the changes to the workflow could be elaborated further. This would bring more depth to the analysis and provide a comprehensive view of the framework's implementation.
It is notable that the findings reveal a significant disparity between the results obtained from manual inspection and those derived from the proposed methodological framework. This indicates the potential of the framework to enhance the quality of Hot Mix Asphalt (HMA) construction processes through effective asset management.
In the introductory section, the authors clearly state that data collection can be executed through various tools, though UAS is specifically chosen for this study. It would be beneficial if the authors could provide a more detailed rationale for preferring UAS over other possible tools, to better convince readers of their choice.

Overall, this is a stimulating and insightful paper that adds value to our understanding of technical applications in HMA construction.

Lastly, I noticed an error on line 310 regarding a figure reference. Kindly take a moment to correct this for the readers' clarity and comprehension.

Author Response

Reviewer 6:

  1. I was very engaged by this paper, which effectively communicates its significant contributions to the field, particularly with its focus on technical applications. The framework introduced in this paper brings substantial novelty by proposing the replacement of manual inspection, data acquisition, and data extraction with Unmanned Aerial Systems (UAS). It is impressive how the authors have delved into the development and outcomes of this framework.

Thank you for your positive feedback on our paper’s contribution. We have carefully considered your comments, aiming to improve the quality of this paper.

  1. However, the discussion seems to emphasize more the technical aspects, while the changes to the workflow could be elaborated further. This would bring more depth to the analysis and provide a comprehensive view of the framework's implementation.

Thank you for your great recommendation. An independent discussion section is added to address this comment, in Lines 396-424 as follows:

“This paper presented a proposed workflow that utilizes UAS for thermal segregation inspection in HMA and auto-mates the data analysis process through Image Processing. The methodology was specifically tested on an MDOT project, involving three key steps: (a) data collection, (b) data processing, and (c) data analysis and result interpretation. The collected data successfully demonstrated the capability to detect temperature ranges, temperature differentials, and segregated areas on the HMA mat.

The main contribution of this research is directed at agencies in the infrastructure domain such as DOTs. This research contributes to the body of knowledge by creating a formal workflow for Automated Pavement Construction Inspection using UAS.

The implementation of the proposed methodology is subject to various limitations that should be taken into account. These limitations include factors such as Federal Aviation Administration (FAA) flying permission constraints, availability of trained personnel, expertise in data processing techniques, and potential influence of weather conditions. The FAA imposes certain conditions and limitations, such as restricting the maximum weight of the UAS to 55 lbs (25 kg), requiring a pilot in command and a visual observer, and mandating that the UAS remains within the visual line of sight. These limitations are comprehensively explained by the authors in [24].

Lastly, this study was conducted as part of the phase three research project titled "Integration of Unmanned Aerial Systems Data Collection into Day-to-Day Usage for Transportation Infrastructure" [24]. In the second phase, the project focused on analyzing the cost and benefits of utilizing UAS for bridge inspections [18]. However, the scope of the third phase research did not include a similar analysis for the application of UAS in HMA thermal inspection. It is important to acknowledge that the workflow proposed in this paper is intended to support labor and streamline the workforce. However, conducting a direct comparison between traditional and automated workflows may not be feasible at present. This is primarily because the cost factor relies on the interplay of supply and demand, and the implementation of the proposed UAS technology lacks information regarding supply and demand dynamics necessary for a comprehensive cost analysis.”

  1. It is notable that the findings reveal a significant disparity between the results obtained from manual inspection and those derived from the proposed methodological framework. This indicates the potential of the framework to enhance the quality of Hot Mix Asphalt (HMA) construction processes through effective asset management.

Yes, as you mentioned there was a significant difference between the manual and UAS-collected results. While the manual method is only dependent on a few samples, the UAS method collects more data points, and provides a temperature profile for the whole mat.

  1. In the introductory section, the authors clearly state that data collection can be executed through various tools, though UAS is specifically chosen for this study. It would be beneficial if the authors could provide a more detailed rationale for preferring UAS over other possible tools, to better convince readers of their choice.

This statement seems to be misleading. We have removed it from the document to make the research scope clear.

  1. Overall, this is a stimulating and insightful paper that adds value to our understanding of technical applications in HMA construction.

We appreciate that you found our paper insightful. This positive feedback serves as a great motivation for us.

  1. Lastly, I noticed an error on line 310 regarding a figure reference. Kindly take a moment to correct this for the readers' clarity and comprehension.

We have revised the cross-reference for “Figure 14”. It looks like an internal error on Word Program, since the reference was initially visible on our system.

Round 2

Reviewer 2 Report

Most of the comments were conducted and addressed in a timely and professional manner, providing helpful insights into the topic at hand. Therefore I agree to accept this work. Thank you for the clarification.

Author Response

Most of the comments were conducted and addressed in a timely and professional manner, providing helpful insights into the topic at hand. Therefore, I agree to accept this work. Thank you for the clarification.

Thank you for your positive feedback. We appreciate your recognition of our prompt and professional handling of the comments, as well as the valuable insights provided on the topic. 

Reviewer 3 Report

Line 118: "While several recent studies [17, 18, 19, 20, 21, 22, 23, 24, 25, 26, 27, and 28]" > This should be "[17-28]"

Issue #4 still needs more work. Any scientific publication must clearly differentiate between research methods and outputs.

Issue #5: A future comparison between traditional and automated workflows may not necessarily be based on costs but primarily on task completion efficiency (usability, data accuracy, interoperability, etc.). The usefulness of technology to accomplish a task comes before its costs. It is recommended to include this point in the Discussion section.

Issue #6: It is important to discuss how the results of the current study compare to or expand upon previous research findings, such as those of Du et al. [23]. This kind of discussion is expected in a scientific paper and can be included in the newly created Discussion section.

Issue #7: Further automating the workflow could be mentioned as another future research direction in the Future Research section.

Author Response

Reviewer 3:

  1. Line 118: "While several recent studies [17, 18, 19, 20, 21, 22, 23, 24, 25, 26, 27, and 28]" > This should be "[17-28]"

Thank you for this comment. This issue is addressed in Line 118, and the test is changed to [17-28].

  1. Issue #4 still needs more work. Any scientific publication must clearly differentiate between research methods and outputs.

Thank you for this comment. The methodology section describes the framework developed in three steps of a) data collection, b) data processing, and c) data analysis. In order to facilitate comprehension for readers without a background in Computer Vision, the authors have made a deliberate effort to incorporate a real-world project example within the methodology. This serves to simplify the various steps involved and provide a practical illustration of Image Processing and Computer Vision concepts. For example, “Figure 5 Grayscale Image from Regular RJPG” provides an example of converting an RGB image to a grayscale image. Another example is “Figure 7 Intensity-Temperature Relationship” where the unifying process of scales is defined using an example. These steps are not the final output of the methodology though. Equations 2-4 and the discussion section are the final output of the project.

  1. Issue #5: A future comparison between traditional and automated workflows may not necessarily be based on costs but primarily on task completion efficiency (usability, data accuracy, interoperability, etc.). The usefulness of technology to accomplish a task comes before its costs. It is recommended to include this point in the Discussion section.

Thank you for your comment. This is addressed in Lines 439 – 441 as:
“Lastly, the effectiveness of the proposed methodology will be examined through the analysis of multiple projects. This evaluation aims to unveil the automated workflow's contributions in terms of usability, accuracy, and interoperability.”

  1. Issue #6: It is important to discuss how the results of the current study compare to or expand upon previous research findings, such as those of Du et al. [23]. This kind of discussion is expected in a scientific paper and can be included in the newly created Discussion section.

Thank you for this comment. Due et al. (2021) have conducted a review on the state of art in image processing and have identified the areas of application. They did not propose any methodology for thermal segregation detection.

  1. Issue #7: Further automating the workflow could be mentioned as another future research direction in the Future Research section.

This comment is addressed in Lines 434 as follows:

“In addition, the workflow illustrated in this paper can be fully automated and implemented on site, and in a near real-time manner, in the presence of appropriate technical hardware.”

Reviewer 5 Report

Thank you for addressing my comments. No further comments from me.

Author Response

We appreciate your review. We are glad we were able to address your comments.